# Cumulative fluid balance predicts mortality and increases time on mechanical ventilation in ARDS patients: An observational cohort study

Niels van Mourik[1,2], Hennie A. Metske[1], Jorrit J. Hofstra[2], Jan M. Binnekade[2], Bart F. Geerts[1], Marcus J. Schultz[2], Alexander P. J. Vlaar[2]*

1 Department of Anaesthesiology, Amsterdam UMC, location AMC, University of Amsterdam, Amsterdam, The Netherlands, 2 Department of Intensive Care Medicine, Amsterdam UMC, location AMC, University of Amsterdam, Amsterdam, The Netherlands

* a.p.vlaar@amc.uva.nl

**Data Availability Statement:** All relevant data are within the paper and its Supporting Information files.

## Abstract

### Introduction

Acute respiratory distress syndrome (ARDS) is characterized by acute, diffuse, inflammatory lung injury leading to increased pulmonary vascular permeability, pulmonary oedema and loss of aerated tissue. Previous literature showed that restrictive fluid therapy in ARDS shortens time on mechanical ventilation and length of ICU-stay. However, the effect of intravenous fluid use on mortality remains uncertain. We investigated the relationship between cumulative fluid balance (FB), time on mechanical ventilation and mortality in ARDS patients.

### Materials and methods

Retrospective observational study. Patients were divided in four cohorts based on cumulative FB on day 7 of ICU-admission: ≤0 L (Group I); 0–3.5 L (Group II); 3.5–8 L (Group III) and ≥8 L (Group IV). In addition, we used cumulative FB on day 7 as continuum as a predictor of mortality. Primary outcomes were 28-day mortality and ventilator-free days. Secondary outcomes were 90-day mortality and ICU length of stay.

### Results

Six hundred ARDS patients were included, of whom 156 (26%) died within 28 days. Patients with a higher cumulative FB on day 7 had a longer length of ICU-stay and fewer ventilator-free days on day 28. Furthermore, after adjusting for severity of illness, a higher cumulative FB was associated with 28-day mortality (Group II, adjusted OR (aOR) 2.1 [1.0–4.6], p = 0.045; Group III, aOR 3.3 [1.7–7.2], p = 0.001; Group IV, aOR 7.9 [4.0–16.8], p<0.001). Using restricted cubic splines, a non-linear dose-response relationship between cumulative FB and probability of death at day 28 was found; where a more positive FB predicted mortality and a negative FB showed a trend towards survival.

**Funding:** The authors received no specific funding for this work.

**Competing interests:** The authors have declared that no competing interests exist.

## Conclusions

A higher cumulative fluid balance is independently associated with increased risk of death, longer time on mechanical ventilation and longer length of ICU-stay in patients with ARDS. This underlines the importance of implementing restrictive fluid therapy in ARDS patients.

## Introduction

Acute respiratory distress syndrome (ARDS) occurs in around 10% of patients admitted to the intensive care unit (ICU) and is still heavily associated with mortality. ARDS is characterized by acute, diffuse, inflammatory lung injury leading to increased pulmonary vascular permeability, pulmonary oedema and loss of aerated tissue [1]. Despite recent advances in the management of ARDS, mortality rates are still reported as high as 40% [2]. Moreover, it was previously demonstrated that a prolonged time on mechanical ventilation in patients with ARDS increases the risk of ventilator-induced injury [3].

Underlying causes of ARDS include sepsis, pneumonia, direct trauma, pancreatitis, blood transfusion and burns [4]. Fluid resuscitation is a mainstay and life-saving intervention in most of these causes. However, a protracted positive fluid balance is also associated with adverse outcome in the critically ill [5, 6]. This may be of even greater importance in patients with ARDS, as pulmonary oedema is one of the key clinical features in this syndrome. Accumulated fluids may deteriorate the patient's clinical condition, resulting in adverse outcome. Indeed, an interventional study showed that a conservative versus a liberal fluid therapy approach significantly decreased time on mechanical ventilation, whereas the effect on mortality remained uncertain [7]. A few observational studies did show a negative association between fluid balance and mortality in specific subsets of patients with ARDS [8–10]. Our study focused on a large set of patients with ARDS, divided into multiple categories based on the height of cumulative fluid balance. Moreover, we looked into the relationship between cumulative fluid balance as a continuum and probability of death.

We aimed to investigate the association between cumulative fluid balance, mortality and ventilator-free days in a large cohort of patients with ARDS admitted to the ICU. We hypothesized that there is a negative association between increased cumulative fluid balance and adverse outcome in patients with ARDS.

## Materials and methods

This cohort study was a retrospective observational study performed on data collected for a previously published study [11]. No separate Medical Ethics Committee approval was required. The study was performed in a 30-bed, closed format, mixed medical-surgical ICU in a tertiary referral hospital (Amsterdam UMC, location AMC, Amsterdam, The Netherlands). Patients were under direct care of a team of intensive care physicians, subspecialty fellows and residents.

Using an electronic patient data monitoring system, patients admitted to the ICU between November 1, 2004 and October 1, 2007 were screened on ARDS and included in the study. ARDS was defined following the Berlin Definition, i.e. acute onset of $PaO_2/FiO_2 \leq 300$ mmHg, bilateral opacities consistent with pulmonary oedema on radiographic imaging, requirement for positive pressure ventilation via endotracheal tube and not fully explained by cardiac failure or fluid overload [1].

Data on patient demographics, disease severity, comorbidities, probable causes of ARDS, vasopressor/inotropics use, diuretics, continuous veno-venous hemofiltration (CVVH) use,

time on mechanical ventilation, length of hospital and ICU stay, 28-day and 90-day mortality and daily and cumulative fluid balances were collected. Cumulative fluid balance was calculated by total fluid input minus total fluid output on a certain day of ICU admission. Insensible fluid losses like transepidermal diffusion or evaporative water loss from the respiratory tract were not routinely measured and not taken into account. Data on type of fluids used was not available.

After data collection, patients were divided in 4 groups based on cumulative fluid balance on day 7 of ICU admission: Group I ($\leq 0$ L); Group II ($> 0$ L—$< 3.5$ L); Group III ($\geq 3.5$L- $<$ 8 L) and Group IV ($\geq 8$ L). In addition, we performed analyses using cumulative fluid balance as continuum as a predictor of outcome. We chose cumulative fluid balance on day 7 to make our results more readily comparable to previously performed studies [7]. In patients who died or were discharged prior to day 7, cumulative fluid balance at the time of disposition was carried forward to day 7. Our primary outcome measures were 28-day mortality and ventilator-free days, where death was penalized as zero ventilator free days. Our secondary outcome measures were 90-day mortality and ICU length of stay.

As the FACTT trial was published during patient inclusion, we also investigated whether there was a difference in cumulative fluid balance in patients included either before or after time of publication [7].

## Statistical analyses

Continuous normally distributed variables were expressed as means with standard deviations or when not normally distributed as medians with interquartile ranges. Categorical variables were expressed as frequencies and percentages and Pearson's Chi-squared test was used to determine differences between groups. No imputation was used to estimate missing data. Analyses were based on all available data with numbers of available observations reported. To determine differences in non-parametric continuous variables, a Kruskal Wallis ANOVA and separate Mann–Whitney U tests were performed. The goal of the primary analysis was to quantify the effect of the cumulative fluid balance on day 7 after ICU admission on mortality, controlling for other variables. We performed binary logistic regression analysis, where Group I was used as reference group. Exploration of confounding was considered methodologically relevant. We first focussed on the crude (uncorrected) effect of fluid balance (independent variable) on mortality (dependent variable). As severity of illness is known to be associated with increased cumulative fluid balances, we added the Acute Physiology And Chronic Health Evaluation II (APACHE II) score as covariate to the model anyhow. Confounding was defined as $\geq 10\%$ change in the estimated measure of association (i.e. odds ratio, OR) of the central determinant (fluid balance) as a consequence of adding a covariate. CVVH and diuretics use, sepsis and gender were considered to be potential confounders and were left in the model when meeting the previous definition. Moreover, we performed restricted cubic splines analyses (after checking for non-linearity), looking at cumulative fluid balance on day 7 as a continuous predictor and death at day 28 as outcome, both unadjusted and adjusted for severity of illness. The restricted cubic spline curves had four equally distributed knots. We considered statistical significance to be at p 0.05. When appropriate, statistical uncertainty was expressed by 95% confidence intervals. Analyses were performed using R in RStudio, version 3.6.0.

## Results

### Demographics

Six hundred ARDS patients were included in this study, of whom 156 (26%) died within 28 days. Non-survivors had higher cumulative fluid balances than survivors in the first week of

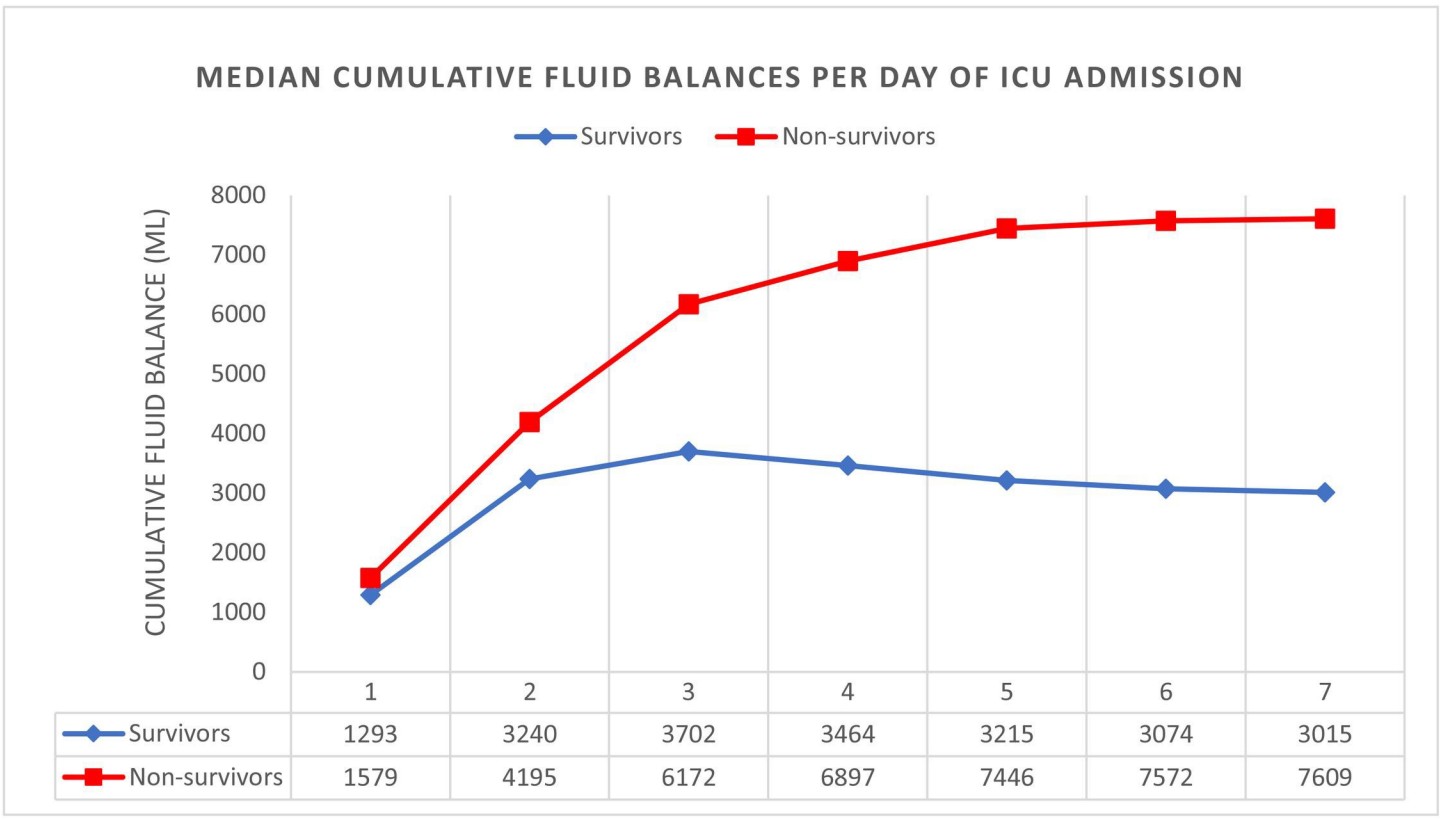

**Fig 1. Median cumulative fluid balance in the first 7 days of ICU admission in survivors and non-survivors.** Survival was based on 28-day mortality. Survivors are marked blue, non-survivors are marked red. On the Y-axis the cumulative fluid balance in mL is found. On the X-axis days of ICU admission with corresponding median cumulative fluid balances (mL) for survivors and non-survivors.

ICU admission (Fig 1). No statistically significant difference was found in median cumulative fluid balance on day 7 before (n = 344) or after (n = 256) the FACTT trial: 4168 [778–9363] mL vs. 3931 [491–7627] mL, p = 0.195. The median $PaO_2/FiO_2$ ratio on diagnosis was 175 mmHg. Median length of ICU stay was 7 [4–13] days. Furthermore, the median age was 62 [49–73] years and the median APACHE II score was 17 [13–23]. The most common cause of ARDS was pneumonia (18.5%), followed by sepsis (16.8%). The prevalence of sepsis was higher in patients with a relatively high cumulative fluid balance. CVVH was more prevalent in patients with a high cumulative fluid balance, whereas furosemide use was similar across all groups. Patient characteristics and outcome measures for all patients, survivors and non-survivors are presented in Table 1. Patient characteristics for patients divided in fluid balance groups are presented in Table 2.

### Outcome measures

Outcome measures are presented in Table 3. We found a significantly longer length of ICU-stay in patients with a higher cumulative fluid balance on day 7 of ICU admission. Moreover, a more positive cumulative fluid balance was associated with fewer ventilator-free days at 28 days.

Death at day 28 and day 90 were significantly more prevalent in patients with a higher cumulative fluid balance. Upon logistic regression analysis, a higher cumulative fluid balance at day 7 was associated with 28-day mortality (Group II crude Odds Ratio (OR) 2.4 [1.2–5.2],

**Table 1. Demographic and clinical data of all patients, survivors and non-survivors.**

| | All patients N = 600 | Survivors N = 444 | Non-survivors N = 156 | P-value |
|---|---|---|---|---|
| Age–years | 62.0 [49.0–73.0] | 60.0 [47.0–72.0] | 67.0 [55.0–76.0] | *0.001* |
| Male–n (%) | 414 (69.0) | 314 (70.7) | 100 (64.1) | 0.151 |
| **Severity of illness** | | | | |
| APACHE II score | 17.0 [13.0–23.0] | 16.0 [12.0–22.0] | 21.0 [16.0–26.0] | *<0.001* |
| **ARDS Severity–n/total n (%)[a]** | | | | 0.093 |
| Mild | 211/595 (35.5) | 149/440 (33.9) | 62/155 (40.0) | |
| Moderate | 334/595 (56.1) | 258/440 (58.6) | 76/155 (49.0) | |
| Severe | 50/595 (8.4) | 33/440 (7.5) | 17/155 (11.0) | |
| **Mortality and length of stay** | | | | |
| 28-day mortality–n (%) | 156 (26.0) | 0 (0) | 156 (100) | *<0.001* |
| 90-day mortality–n (%) | 202 (33.7) | 46 (10.4) | 156 (100) | *<0.001* |
| ICU mortality–n (%) | 110 (18.3) | 4 (0.9) | 106 (67.9) | *<0.001* |
| ICU length of stay–days | 6.7 [3.9–12.6] | 6.3 [3.9–12.6] | 7.7 [4.3–127] | 0.533 |
| Hospital length of stay–days | 20.0 [10.9–38.0] | 24.4 [12.6–43.2] | 13.0 [7.0–23.9] | *<0.001* |
| **Respiratory variables** | | | | |
| PaO$_2$/FiO$_2$ on diagnosis–mmHg | 175 [132–229] | 171 [134–227] | 180 [126–232] | 0.770 |
| Duration of mechanical ventilation–hrs | 101 [51.0–228] | 89.5 [46.5–214] | 136 [70.8–242] | *0.002* |
| Ventilator free-days on day 28 | 24.2 [19.0–26.0] | 24.3 [19.1–26.1] | 0.00 [0.00–0.00] | *<0.001* |
| Weaned off ventilator day 28 –n (%) | 422 (70.3) | 421 (94.8) | 0 (0) | *<0.001* |
| Protective ventilation [12]–n/total n (%)[b] | 294/574 (51.2) | 207/430 (48.1) | 87/144 (60.4) | *0.008* |
| **Fluid therapy and organ support** | | | | |
| Cumulative FB day 7 –mL | 3994 [620–8467] | 3015 [-62.25–6443] | 7609 [3116–13275] | *<0.001* |
| CVVH–n (%)[c] | 70 (11.1) | 37 (8.33) | 33 (21.2) | *<0.001* |
| Furosemide–n(%) [c] | 313 (52.2) | 232 (52.3) | 81 (51.9) | 1.000 |
| Vasopressors/inotropics at onset ARDS–n/total n (%) | 348/597 (58.3) | 242/442 (54.8) | 106/155 (67.9) | *0.006* |
| **ICU admission category–n (%)** | | | | |
| Medical | 349 (58.2) | 240 (54.1) | 109 (69.9) | *<0.001* |
| Elective surgery | 147 (24.5) | 128 (28.8) | 19 (12.2) | *<0.001* |
| Emergency surgery | 104 (17.3) | 76 (17.1) | 28 (17.9) | 0.910 |
| **Probable ARDS causes–n (%)** | | | | |
| Sepsis | 101 (16.8) | 66 (14.9) | 35 (22.4) | *0.040* |
| Pneumonia | 111 (18.5) | 81 (18.2) | 30 (19.2) | 0.878 |
| Aspiration | 30 (5.0) | 22 (5.0) | 8 (5.1) | 1.000 |
| Massive transfusion | 93 (15.5) | 75 (16.9) | 18 (11.5) | 0.144 |
| Near drowning | 3 (0.5) | 0 (0) | 3 (1.9) | *0.017* |
| Pancreatitis | 12 (2.0) | 11 (2.5) | 1 (0.6) | 0.201 |
| Trauma | 42 (7.0) | 36 (8.1) | 6 (3.9) | 0.101 |
| **Comorbidities** | | | | |
| COPD | 66 (11.0) | 44 (9.9) | 22 (14.1) | 0.197 |
| Diabetes | 88 (14.8) | 56 (12.7) | 32 (20.8) | 0.021 |
| Alcohol abuse | 53 (8.8) | 44 (9.9) | 9 (5.8) | 0.160 |
| Auto-immune disease | 40 (6.7) | 25 (5.6) | 15 (9.6) | 0.126 |
| Immune compromised | 62 (10.3) | 39 (8.8) | 23 (14.7) | 0.051 |

Values indicated with n are number of patients. Medians are presented with interquartile ranges between square brackets. Survival is based on 28-day mortality. P-values were calculated for the differences between survivors and non-survivors. FB denotes fluid balance; CVVH, continuous veno-venous hemofiltration; ARDS, Acute Respiratory Distress Syndrome; APACHE, Acute Physiology and Chronic Health Evaluation; COPD, Chronic Obstructive Pulmonary Disease.

[a]Defined using PaO$_2$/FiO$_2$ ratio on diagnosis, following the Berlin Definition [1].

[b]Protective ventilation was defined as 6 mL/kg of predicted body weight.

[c]During the first 31 days of ICU admission.

**Table 2. Demographic and clinical data of patients subdivided in cumulative fluid balance groups.**

| | Group I < 0 L N = 123 | Group II 0–3.5 L N = 177 | Group III 3.5–8L N = 151 | Group IV ≥8 L N = 149 | P-value |
|---|---|---|---|---|---|
| Age–years | 57.0 [48.5–70.0] | 65.0 [54.0–74.0] | 60.0 [45.0–73.0] | 64.0 [49.0–74.0] | 0.075 |
| Male–n (%) | 92 (74.8) | 129 (72.9) | 97 (64.2) | 96 (64.4) | 0.101 |
| **Severity of illness** | | | | | |
| APACHE II score | 15.0 [12.0–22.0]^IV | 18.0 [13.0–23.0] | 17.0 [13.0–22.5] | 19.0 [15.0–24.0]^I | *0.001* |
| **ARDS Severity n/total n (%)[a]** | | | | | 0.117 |
| Mild | 33/119 (27.0) | 63/175 (36.0) | 63/149 (42.3) | 52/149 (34.9) | |
| Moderate | 74/199 (60.7) | 95/175 (54.3) | 78/149 (52.3) | 87/149 (58.4) | |
| Severe | 15/199 (12.3) | 17/175 (9.7) | 8/149 (5.4) | 10/149 (6.7) | |
| **Respiratory variables** | | | | | |
| PaO₂/FiO₂ on diagnosis–mmHg | 164 [126–212]^III | 170 [129–230] | 185 [136–243]^I | 180 [137–225] | *0.047* |
| Protective ventilation [12]–n/total n (%)[b] | 59/119 (49.6) | 90/171 (52.6) | 68/145 (46.8) | 77/139 (55.4) | 0.551 |
| **Fluid therapy and organ support** | | | | | |
| Cumulative FB day 7 –mL | -2107.00 [-3773.00--1133.50]^II, III, IV | 2034 [1099–2924]^I,III, IV | 5737 [4822–6838]^I,II, IV | 12696 [9925–17475]^I,II, III | *<0.001* |
| CVVH–n (%)[c] | 5 (4.1)^IV | 7 (3.9)^IV | 13 (8.6)^IV | 45 (30.2)^I,II,III | *<0.001* |
| Furosemide–n(%) [c] | 60 (48.8) | 97 (54.8) | 72 (47.7) | 84 (56.4) | 0.343 |
| Vasopressors/inotropics at onset ARDS–n/total n (%) | 51/123 (41.5)^III, IV | 90/176 (51.1)^IV | 85/150 (56.7)^I, IV | 122/148 (82.4)^I,II,III | *<0.001* |
| **ICU admission category–n (%)** | | | | | |
| Medical | 70 (56.9) | 104 (58.8) | 91 (60.3) | 84 (56.4) | 0.902 |
| Elective surgery | 33 (26.8) | 46 (26.0) | 35 (23.2) | 33 (22.1) | 0.762 |
| Emergency surgery | 20 (16.3) | 27 (15.3) | 25 (16.6) | 32 (21.5) | 0.479 |
| **Probable ARDS causes–n (%)** | | | | | |
| Sepsis | 10 (8.1)^III, IV | 18 (10.2)^IV | 28 (18.5)^I,IV | 45 (30.2)^I,II,III | *<0.001* |
| Pneumonia | 21 (17.1) | 36 (20.3) | 32 (21.2) | 22 (14.8) | 0.446 |
| Aspiration | 9 (7.3) | 8 (4.5) | 5 (3.3) | 8 (5.4) | 0.489 |
| Massive transfusion | 16 (13.0) | 25 (14.1) | 22 (14.6) | 30 (20.1) | 0.336 |
| Near drowning | 0 (0) | 0 (0) | (0) | 2 (2.1) | *0.024* |
| Pancreatitis | 6 (4.9)^II | 0 (0)^I | 4 (2.7) | 2 (1.34) | *0.011* |
| Trauma | 6 (4.9) | 9 (5.1) | 19 (12.6) | 8 (5.4) | *0.021* |
| **Comorbidities** | | | | | |
| COPD | 18 (14.6) | 19 (10.7) | 16 (10.6) | 13 (8.7) | 0.478 |
| Diabetes | 21 (17.1) | 32 (18.3) | 18 (11.9) | 17 (11.6) | 0.220 |
| Alcohol abuse | 9 (7.3) | 19 (10.7) | 14 (9.3) | 11 (7.4) | 0.666 |
| Auto-immune disease | 5 (4.1) | 9 (5.1) | 13 (8.6) | 13 (8.7) | 0.264 |
| Immune compromised | 10 (8.1) | 15 (8.5) | 18 (11.9) | 19 (12.8) | 0.448 |

Values indicated with n are number of patients. Medians are presented with interquartile ranges between square brackets. Significantly differing groups (p<0.05) are displayed in superscript. FB denotes fluid balance; CVVH, continuous veno-venous hemofiltration; ARDS, Acute Respiratory Distress Syndrome; APACHE, Acute Physiology and Chronic Health Evaluation; COPD, Chronic Obstructive Pulmonary Disease.

[a]Defined using PaO₂/FiO₂ ratio on diagnosis, following the Berlin Definition [1].

[b]Protective ventilation was defined as 6 mL/kg of predicted body weight.

[c]During the first 31 days of ICU admission.

p = 0.017; Group III OR 3.7 [1.8–7.8], p<0.001; Group IV OR 9.3 [4.8–19.5], p<0.001). This finding persisted after adjusting for severity of illness by APACHE II score (Group II adjusted Odds Ratio (aOR) 2.1 [1.0–4.6], p = 0.045; Group III, aOR 3.3 [1.7–7.2], p = 0.001; Group IV

**Table 3. Outcome measures for patients subdivided in cumulative fluid balance groups.**

| | Group I < 0 L N = 123 | Group II 0–3.5 L N = 177 | Group III 3.5–8L N = 151 | Group IV ≥8 L N = 149 | P-value |
|---|---|---|---|---|---|
| **Mortality–n (%)** | | | | | |
| 28-day mortality | 11 (8.9)[II,III,IV] | 34 (19.2)[I, IV] | 40 (26.5)[I, IV] | 71 (47.7)[I,II,III] | <0.001 |
| 90-day mortality | 16 (13.0)[II,III,IV] | 50 (28.2)[I, IV] | 52 (34.4)[I, IV] | 84 (56.4)[I,II,III] | <0.001 |
| ICU mortality | 5 (4.1)[III, IV] | 15 (8.5)[III, IV] | 33 (21.9)[I, II, IV] | 57 (38.3)[I, II, III] | <0.001 |
| **Ventilation** | | | | | |
| Ventilator-free days on day 28 | 25.1 [20.8–26.6][II, III, IV] | 24.3 [10.4–26.0][I, III, IV] | 19.2 [0.00–25.1][I, II, IV] | 0.00 [0.00–19.5][I, II, III] | <0.001 |
| Weaned off ventilator by day 28 | 107 (87.0)[III, IV] | 139 (78.5)[IV] | 106 (70.2)[I, IV] | 70 (47.0)[I, II, III] | <0.001 |
| Duration of mechanical ventilation–hours | 62.0 [31.5–136][III, IV] | 80.0 [39.0–136][III, IV] | 132 [61.0–257][I, II, IV] | 189 [101–358][I, II, III] | <0.001 |
| **Length of stay–days** | | | | | |
| ICU length of stay | 5.03 [3.05–8.65][III, IV] | 5.45 [3.49–8.32][III, IV] | 8.39 [4.65–12.8][I, IV] | 11.6 [6.35–20.7][I, II, III] | <0.001 |
| Hospital length of stay | 17.0 [10.0–32.5][IV] | 15.0 [9.00–35.0][IV] | 21.0 [12.0–38.0] | 28.5 [14.0–45.0][I, II] | 0.004 |

Values indicated with n are number of patients. Medians are presented with interquartile ranges between square brackets. Significantly differing groups (p<0.05) are displayed in superscript. ICU denotes Intensive Care Unit.

aOR 7.9 [4.0–16.8], p<0.001). While exploring for confounders using our pre-determined covariates, no other confounders were identified. Similar results were found for 90-day mortality (Group II OR 2.6 [1.4–5.0], p = 0.002, aOR 2.3 [1.3–4.5], p = 0.008; Group III OR 3.5 [1.9–6.7], p<0.001, aOR 3.2 [1.7–6.3], p<0.001; Group IV OR 8.6 [4.8–16.5], p<0.001, aOR 7.4 [4.0–14.3], p<0.001).

Using restricted cubic splines, a non-linear dose-response relationship was found between cumulative fluid balance on day 7 and 28-day mortality. A more positive fluid balance predicted 28-day mortality, whereas there was a trend for negative fluid balance to predict survival. These findings persisted after adjusting for severity of illness (APACHE II). See Figs 2 and 3.

## Discussion

We investigated the association between cumulative fluid balance, mortality and ventilator-free days in a large cohort of ICU patients with ARDS. A more positive cumulative fluid balance on day 7 was associated with an increased risk of death, even after adjusting for severity of illness (APACHE II score). Moreover, a higher positive cumulative fluid balance was associated with fewer ventilator-free days and an increased length of ICU-stay. We found a non-linear relationship between 28-day mortality and cumulative fluid balance on day 7. A more positive cumulative fluid balance predicted death, whereas a negative fluid balance showed a trend towards survival. Found association could be described as a dose-response relationship, as patients with higher cumulative fluid balances had worse outcome.

CVVH was associated with 28-day mortality and higher cumulative fluid balances. CVVH is an indicator of acute kidney injury (AKI), which contributes to the accumulation of fluids. Nonetheless, fluid accumulation itself can also result in AKI [13]. Unfortunately, due to the observational nature of our study it remains uncertain whether AKI contributed to the accumulation of fluid or vice versa. Similar to CVVH, sepsis was also associated with 28-day mortality and with higher cumulative fluid balances. During hemodynamic resuscitation intravascular volume is often depleted rapidly, due to capillary leakage and vasoplegia, resulting in massive fluid loading. Indeed, sepsis was more prevalent in patients with higher fluid

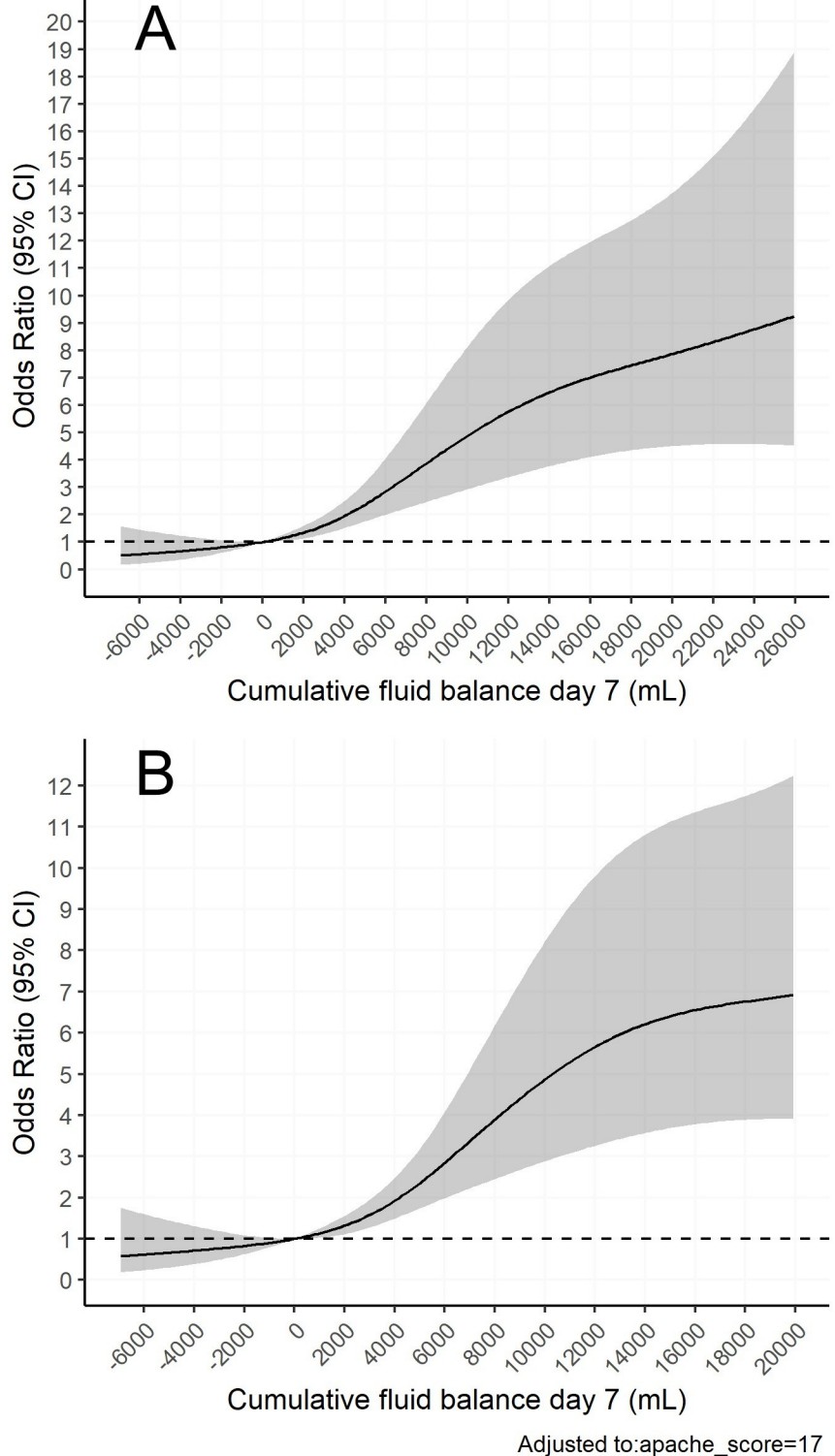

**Fig 2. The relationship between odds of death and cumulative fluid balance.** Restricted cubic splines models. Confidence intervals (95%) are displayed in grey. Neutral fluid balance (0 mL) was used as reference (OR 1.0). A) Unadjusted odds ratios for 28-day mortality on the y-axis. On the x-axis cumulative fluid balance on day 7 in mL. B) Odds ratios for 28-day mortality on the y-axis, adjusted for severity of illness by APACHE II score. On the x-axis cumulative fluid balance on day 7 in mL.

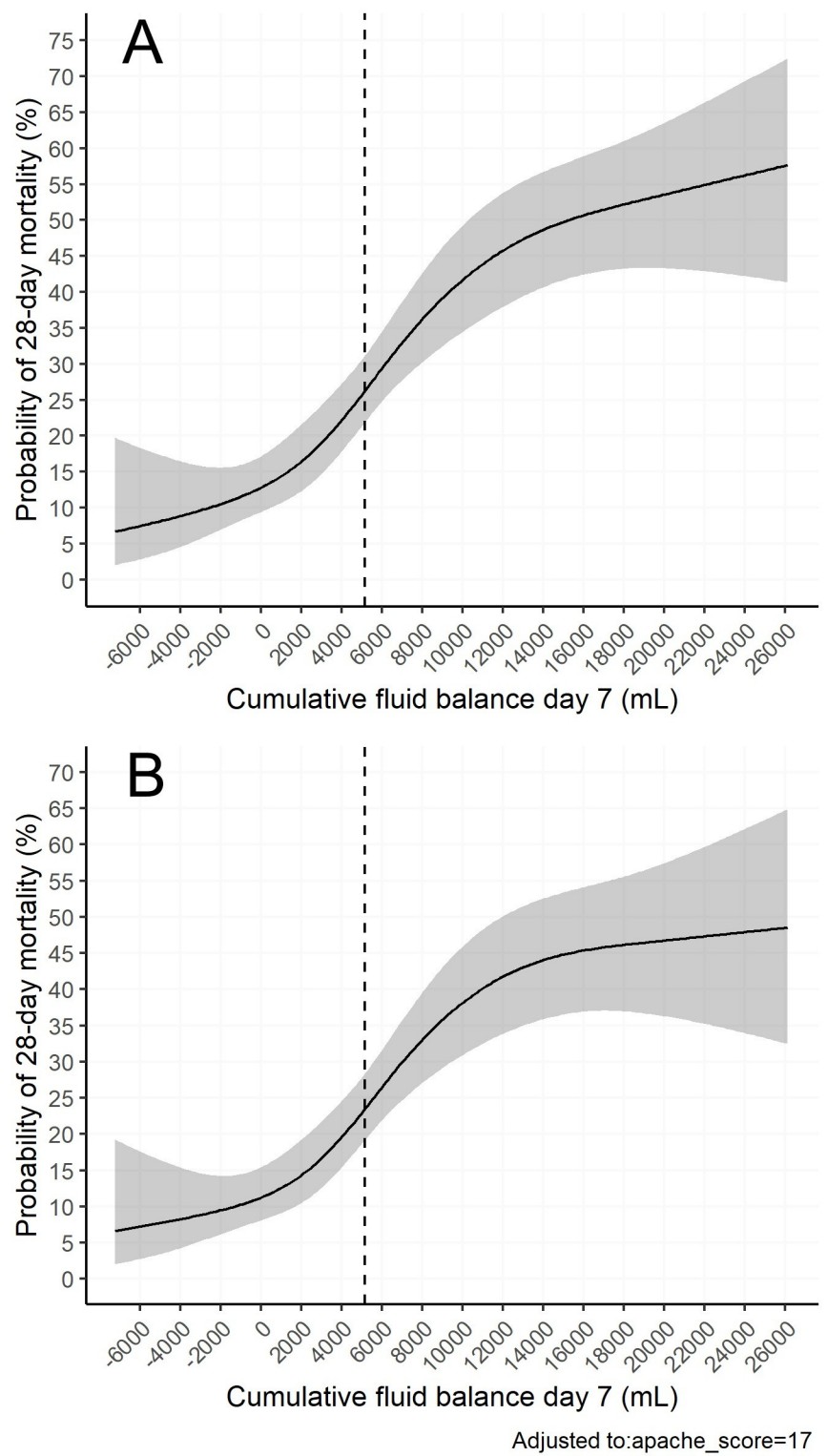

**Fig 3. The relationship between probability of death and cumulative fluid balance.** Restricted cubic splines models. Confidence intervals (95%) are displayed in grey. The average cumulative fluid balance (5139 mL) is marked with the dashed line. A) Unadjusted probability of 28-day mortality on the y-axis. On the x-axis cumulative fluid balance on day 7 in mL. B) Probability of 28-day mortality on the y-axis, adjusted for severity of illness by APACHE II score. On the x-axis cumulative fluid balance on day 7 in mL.

balances. Multivariate analysis did not show CVVH, nor sepsis to be confounding in the prediction of mortality by cumulative fluid balance.

The 2006 Fluids and Catheters Treatment Trial (FACTT) investigated the effects of fluid restriction on outcome in patients with acute lung injury (ALI) [7]. In this randomized clinical trial, a conservative versus a liberal fluid management approach was used for 7 days in 1000 ALI patients. Patients in the conservative-strategy group had significantly lower cumulative fluid balances on day 7 in comparison to the liberal-strategy group (respectively -136 (Standard Error (SE): 491 mL) vs 6992 (SE: 502 mL). No significant difference was found in 60-day mortality. Yet, conservative fluid strategy improved lung function, shortened duration of mechanical ventilation and shortened ICU length of stay. Moreover, a retrospective review of FACTT showed that a negative fluid balance was associated with lower hospital mortality, more ventilator-free days and a shorter ICU length of stay [14]. These results are comparable to our study. Additionally, we showed that positive cumulative fluid balance was a predictor of 28-day and 90-day mortality. The mortality rates in our study are similar to previous ARDS studies [15]. Studies in other subsets of critically ill patients showed similar results. In patients with sepsis (often subject to ARDS) fluid balance was also predictive of mortality [5, 16–18]. Moreover, positive fluid balances were associated with a longer length of stay and a prolonged time on mechanical ventilation.

The mechanisms involved with the negative impact of fluid balance on the outcome of ARDS patients are still not fully elucidated. One explanation is that in patients with ARDS, pulmonary oedema may increase when receiving too much fluid. An increase in pulmonary oedema causes a reduction of pulmonary compliance and an increase in respiratory work. Alveolar oedema may cause intrapulmonary shunting resulting in hypoxemia. Pulmonary oedema is one of the key determinants of pulmonary arterial hypertension, not only by causing hypoxemia but also due to pulmonary vascular compression [19]. Hereby, accumulated fluids may prolong time on mechanical ventilation and increase mortality rates. A prolonged time on mechanical ventilation may result in ventilator-induced injury and vice versa [3]. Patients with ARDS may be more susceptible to ventilator-induced injury as their respiratory system is already affected. Actively reducing accumulated fluids after initial resuscitation might improve outcome.

Our study has its limitations. The retrospective and observational study design limits the possibility to determine a cause-effect relationship between fluid balance and outcome. Even though we have a large patient cohort in which we used multivariate analysis to correct for potential confounding, residual confounding may still exist. Moreover, previous observational studies already showed that a positive fluid balance is associated with mortality in patients with ARDS due to a variety of causes, secondary to septic shock or concomitant with acute kidney injury [8–10, 20]. Our study adds to this body of evidence. In addition to previous studies, we visually displayed a non-linear dose-response relationship between cumulative fluid balance and mortality; with more positive fluid balances predicting mortality, while negative fluid balances seemed to protect against mortality. It is relevant to note that our data are relatively old. The first patient was entered in our database in 2004 and the last patient was entered in 2007. In this period several potential changes in clinical practice could have occurred impacting fluid balance and overall mortality. However, we studied the influence of time of entry in the database and fluid balance and did not find any significant trend over time that would indicate this. This suggests that restrictive fluid therapy was not yet fully implemented in our ICU, which could contribute to the mortality rate. Moreover, no data on type of fluids administered were collected. It would have been interesting to look into the effects of fluid type on outcome. A meta-analysis showed that colloid therapy with albumin improved oxygenation but did not affect mortality when compared to saline. Further research on this topic is needed [21]. Also, it

would be interesting to look into the effects of fluid balance on long-term outcomes. A previous study showed that a conservative fluid management strategy in patients with ARDS is a potential risk factor for long-term cognitive impairment [22]. Unfortunately, we did not collect long-term outcomes.

Our study underlines the importance of implementing conservative fluid balance approaches in patients with ARDS. Nevertheless, positive fluid balances may not always be avoidable in patients with ARDS, as some conditions (e.g. septic shock) are dependent on initial aggressive fluid therapy for resuscitation. A recent retrospective study in adults requiring invasive mechanical ventilation showed that a negative fluid balance achieved with de-resuscitation is associated with improved patient outcome [23]. Whether a de-resuscitative approach in ARDS patients may reverse negative outcome associated with cumulative fluid balance needs further research. Prospective clinical trials are warranted to further investigate de-resuscitation in ARDS.

## Conclusions

We found that a higher cumulative fluid balance is independently associated with increased risk of death, longer time on mechanical ventilation and a longer length of ICU-stay in patients with ARDS. This underlines the importance of implementing restrictive fluid therapy protocols in ARDS patients.

## Supporting information

**S1 Dataset. Raw censored dataset.** Privacy sensitive patient data were excluded.
(XLSX)

## Author Contributions

**Conceptualization:** Hennie A. Metske, Jorrit J. Hofstra, Jan M. Binnekade, Marcus J. Schultz, Alexander P. J. Vlaar.

**Formal analysis:** Niels van Mourik, Jan M. Binnekade.

**Investigation:** Niels van Mourik, Hennie A. Metske, Jorrit J. Hofstra, Jan M. Binnekade, Bart F. Geerts, Marcus J. Schultz, Alexander P. J. Vlaar.

**Methodology:** Niels van Mourik, Hennie A. Metske, Jorrit J. Hofstra, Jan M. Binnekade, Marcus J. Schultz, Alexander P. J. Vlaar.

**Supervision:** Bart F. Geerts, Alexander P. J. Vlaar.

**Visualization:** Niels van Mourik.

**Writing – original draft:** Niels van Mourik.

**Writing – review & editing:** Niels van Mourik, Hennie A. Metske, Jorrit J. Hofstra, Jan M. Binnekade, Bart F. Geerts, Marcus J. Schultz, Alexander P. J. Vlaar.

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
