## [Decision Letter · Decision Letter 0]

7 Oct 2019

PONE-D-19-22932

Cumulative fluid balance predicts mortality and increases time on mechanical ventilation in ARDS patients: an observational cohort study

PLOS ONE

Dear Dr Vlaar,

Thank you for submitting your manuscript to PLOS ONE. After careful consideration, we feel that it has merit but does not fully meet PLOS ONE’s publication criteria as it currently stands. Therefore, we invite you to submit a revised version of the manuscript that addresses the points raised during the review process.

We would appreciate receiving your revised manuscript by Nov 21 2019 11:59PM. To enhance the reproducibility of your results, we recommend that if applicable you deposit your laboratory protocols in protocols.io, where a protocol can be assigned its own identifier (DOI) such that it can be cited independently in the future. For instructions see: http://journals.plos.org/plosone/s/submission-guidelines#loc-laboratory-protocols

We look forward to receiving your revised manuscript.

Kind regards,

Chiara Lazzeri

Academic Editor

PLOS ONE

Journal Requirements:

Journal Requirements

2. Please upload a new copy of Figures 2 and 3 as the detail is not clear. Please follow the link for more information: http://blogs.PLOS.org/everyone/2011/05/10/how-to-check-your-manuscript-image-quality-in-editorial-manager/

Additional Editor Comments (if provided):

Reviewers' comments:

Reviewer's Responses to Questions

**Comments to the Author**

1. Is the manuscript technically sound, and do the data support the conclusions?

Reviewer #1: Yes

Reviewer #2: Yes

2. Has the statistical analysis been performed appropriately and rigorously? 

Reviewer #1: Yes

Reviewer #2: Yes

3. Have the authors made all data underlying the findings in their manuscript fully available?

Reviewer #1: No

Reviewer #2: Yes

4. Is the manuscript presented in an intelligible fashion and written in standard English?

Reviewer #1: Yes

Reviewer #2: Yes

5. Review Comments to the Author

Reviewer #1: This is a good well-conceived and well-written paper on a hot topic where prospective controlled studies are scarce. Conclusions are reasonable regarding results. Although previous papers have published data consistent with the results observed in the present paper, only the Wiedemann paper is a prospective randomized study while all other papers are retrospective studies carried on a relatively small amount of patients.

I have one major and two minor point

Major: The authors state that "all relevant data are within the manuscript". Considering that individual data are needed, they are not available in the manuscript. So, the authors must indicate how to have access to the full data.

Minor: Data are relatively old, more than 15 years for the oldest patients. Presumably, standard of care has moved today towards a more conservative approach about fluids. This could change the overall mortality. This point should be briefly discussed and mentioned in the limitations of the paper.

In table 2, how is defined protective ventilation?

Reviewer #2: A retrospective analysis of a big but old database showing that FB is related to the mortality. The main problem is that FB is related to the severity of disease or to the physicians, in other words in septic patients high vasopressor is related to the outcome. But we did not discourage the use of vasopress. So the FB is the egg or the chicken ?

Thus a correct FB, a correct circulating volemia shoul be applied

Introduction

One of the main problem is how FB was calculated

How was the accuracy of FB ??

Could you comment the FB and the severity of ARDS and of the underlying disease

Could you divide the patients according to the severity of ARDS

How was the protocols for fluid management

Did the protocol changed through the years ?

Methods

Your computation of FB does not take into account the gastrointestinal losses, sweat …..

How did you manage the patients who died before day 7

Why did you not assess the ICU outcome

Could you please insert more data on MV settings at admission and during the ICU stay

Did you have any data on acid base status

In tables you present only P value, but the statistical difference among the groups?

Were ECMO patients ??

Any difference on the type of fluid

Thus I suggest the authors to provide more results and not only the FB

Discussion

Insert and comments the etiology of death

Main problem is not positive or negative FB, is a correct circulating volume and the “rules” for fluid management

Please comment the role of albumin and colloids

6. PLOS authors have the option to publish the peer review history of their article (what does this mean?). If published, this will include your full peer review and any attached files.

Reviewer #1: No

Reviewer #2: No

---

## [Author Response · Author response to Decision Letter 0]

14 Oct 2019

Response to Reviewers

We thank the editor and reviewers for their valuable time, insights and the possibility to improve our manuscript. We hope that we have now sufficiently improved the manuscript. Below you will find our responses. 

Reviewer 1

Comment 1

“Major: The authors state that "all relevant data are within the manuscript". Considering that individual data are needed, they are not available in the manuscript. So, the authors must indicate how to have access to the full data.”

We thank the reviewer for pointing this out. Unfortunately, we were not aware individual patient data were needed. Censored raw patient data are now available in S1 Dataset. 

Comment 2

“Minor: Data are relatively old, more than 15 years for the oldest patients. Presumably, standard of care has moved today towards a more conservative approach about fluids. This could change the overall mortality. This point should be briefly discussed and mentioned in the limitations of the paper.”

We agree on this. The first patient was entered in our database in 2004 and the last patient was entered in 2007. In this period several potential changes in clinical practice could have occurred impacting fluid balance. However, we studied the influence of time of entry in the database and fluid balance and did not find any significant trend over time that would indicate this. We added this to the limitations of the paper, please see page 13, lines 185-191 of the revised manuscript 

Comment 3

“In table 2, how is defined protective ventilation?”

Protective ventilation was defined as 6 mL/kg predicted body weight following the ARDSNet ARMA trial. We added the definition and reference to the table legends. 

Reviewer 2

Comment 1

“One of the main problem is how FB was calculated. How was the accuracy of FB ??”

We agree with the reviewer that the accuracy of measuring is important. 

Fluid balance was calculated based on total fluid input (i.e. all fluids administered enterally and parenterally) and total fluid output (urine, blood loss, stool, vomit, paracentesis, etc). It was not feasible to correct for insensible water losses like transepidermal diffusion or evaporative water loss from the respiratory tract, as we do not routinely measure this. This reflects current medical practice. 

We added a comment to the revised manuscript that we did not take into account insensible fluid losses. Please see page 4, lines 65-67.

Comment 2

“Could you comment the FB and the severity of ARDS and of the underlying disease”

We thank the reviewer for this suggestion. We agree that it would be interesting to look at the severity of ARDS specifically and fluid balance as well. We added the severity of ARDS based on PaO2/FiO2 ratio into the tables (mild – moderate – severe, according to the Berlin criteria). We chose to define severity of illness based on APACHE II score, as was already displayed in the tables.

Comment 3

“Could you divide the patients according to the severity of ARDS”

Please see Comment 2.

Comment 4

“How was the protocols for fluid management. Did the protocol changed through the years ?”

We investigated changes in fluid management by analysing differences in fluid balance before or after the FACTT trial (results, page 6, lines 104-105), which we did not find. This suggests that, even after Wiedemann’s study, restrictive fluid therapy in ARDS was not yet fully implemented in our ICU. 

Whilst we agree with the reviewer that it is interesting to look into the fluid management protocols, we believe that by adding fluid management protocols for all different morbidities the manuscript would become a tedious read. Also and maybe even more important, based on clinical parameters, fluid management may have changed for individual patients during the often unpredictable course of critical illness and may have been subject to the clinician’s own views and preferences. 

Comment 5

“Your computation of FB does not take into account the gastrointestinal losses, sweat …..”

We thank the reviewer for pointing this out. We addressed this above (Comment 1).

Comment 6

“How did you manage the patients who died before day 7”

We agree that it is important to take into account. Patients that died or were discharged before day 7 were addressed in the methods section (original manuscript, page 4, lines 70-71): “In patients who died or were discharged prior to day 7, cumulative fluid balance at the time of disposition was carried forward to day 7.”

Comment 7

“Why did you not assess the ICU outcome”

We thank the reviewer for this suggestion. Assuming with outcome ICU mortality was meant, we added this to the manuscript. 

Comment 8

“Could you please insert more data on MV settings at admission and during the ICU stay”

As we do not wish to publish data twice, we would like to refer to Vlaar et al (Crit Care Med, 2010) for data on ventilatory settings.

Comment 9

“Did you have any data on acid base status”

Unfortunately, we do not have any data on acid-base status.

Comment 10

“In tables you present only P value, but the statistical difference among the groups?”

We agree with the reviewer that it is important to show statistical difference between the different groups separately as well. That is why significantly differing groups were displayed in superscript within the tables. 

Comment 11

“Were ECMO patients ??”

There were no patients on ECMO in the study. 

Comment 12

“Any difference on the type of fluid”

Indeed, it would be interesting to look into the different types of fluids used and outcome. However, as stated in the methods section of the original manuscript (page 4, paragraph 3, line 65): “Data on type of fluids used was not available.”.

Comment 13

“Insert and comments the etiology of death”

No data was available on cause-specific mortality.

Comment 14

“Main problem is not positive or negative FB, is a correct circulating volume and the “rules” for fluid management”

We absolutely agree with the reviewer that there is more to fluid management than just fluid balance. Our study showed that fluid balance is an (independent) predictor of outcome in patients with ARDS. Fluid balance can and probably should be used as an aid in the clinician’s arsenal for fluid management. 

Comment 15

“Please comment the role of albumin and colloids”

As stated above, we agree with the reviewer that this is an interesting point. We added a short section on crystalloids and colloids in ARDS in the discussion. Please see page 13, lines 191-194 of the revised manuscript

---

## [Editor Report · Decision Letter 1]

17 Oct 2019

Cumulative fluid balance predicts mortality and increases time on mechanical ventilation in ARDS patients: an observational cohort study

PONE-D-19-22932R1

Dear Dr. Vlaar,

We are pleased to inform you that your manuscript has been judged scientifically suitable for publication and will be formally accepted for publication once it complies with all outstanding technical requirements.

With kind regards,

Chiara Lazzeri

Academic Editor

PLOS ONE
---

## [Editor Report · Acceptance letter]

21 Oct 2019

PONE-D-19-22932R1 

Cumulative fluid balance predicts mortality and increases time on mechanical ventilation in ARDS patients: an observational cohort study 

Dear Dr. Vlaar:

I am pleased to inform you that your manuscript has been deemed suitable for publication in PLOS ONE. Congratulations! Your manuscript is now with our production department. 

With kind regards,

on behalf of

Dr. Chiara Lazzeri 

Academic Editor

PLOS ONE